# Nutritional Support of Neurodevelopment and Cognitive Function in Infants and Young Children—An Update and Novel Insights

**DOI:** 10.3390/nu13010199

**Published:** 2021-01-10

**Authors:** Kathrin Cohen Kadosh, Leilani Muhardi, Panam Parikh, Melissa Basso, Hamid Jan Jan Mohamed, Titis Prawitasari, Folake Samuel, Guansheng Ma, Jan M. W. Geurts

**Affiliations:** 1School of Psychology, University of Surrey, Guildford GU2 7XH, UK; k.cohenkadosh@surrey.ac.uk (K.C.K.); m.basso@surrey.ac.uk (M.B.); 2FrieslandCampina AMEA, Singapore 039190, Singapore; leilani.muhardi@frieslandcampina.com (L.M.); panam.parikh@icloud.com (P.P.); 3Department of General Psychology, University of Padova, 35131 Padova, Italy; 4Nutrition and Dietetics Programme, School of Health Sciences, Universiti Sains Malaysia, Kubang Kerian 16150, Malaysia; hamidjan@usm.my; 5Nutrition and Metabolic Diseases Working Group, Indonesian Pediatric Society, Jakarta 10310, Indonesia; tprawitasari@yahoo.com; 6Department of Pediatrics, Faculty of Medicine, Universitas Indonesia, Dr. Cipto Mangunkusomo National Referral Hospital Jakarta, Jakarta 10430, Indonesia; 7Department of Human Nutrition, University of Ibadan, Ibadan 200284, Nigeria; samuelfolake@yahoo.co.uk; 8Department of Nutrition and Food Hygiene, School of Public Health, Peking University, 38 Xue Yuan Road, Haidian District, Beijing 100191, China; mags@bjmu.edu.cn; 9Laboratory of Toxicological Research and Risk assessment for Food Safety, Peking University, 38 Xue Yuan Road, Haidian District, Beijing 100191, China; 10FrieslandCampina, 3818 LE Amersfoort, The Netherlands

**Keywords:** brain, neurodevelopment, childhood, protein quality, tyrosine, tryptophan, poly-unsaturated fatty acids, polar lipids, minerals, vitamins, kynurenine, gut-brain axis, prebiotics, probiotics

## Abstract

Proper nutrition is crucial for normal brain and neurocognitive development. Failure to optimize neurodevelopment early in life can have profound long-term implications for both mental health and quality of life. Although the first 1000 days of life represent the most critical period of neurodevelopment, the central and peripheral nervous systems continue to develop and change throughout life. All this time, development and functioning depend on many factors, including adequate nutrition. In this review, we outline the role of nutrients in cognitive, emotional, and neural development in infants and young children with special attention to the emerging roles of polar lipids and high quality (available) protein. Furthermore, we discuss the dynamic nature of the gut-brain axis and the importance of microbial diversity in relation to a variety of outcomes, including brain maturation/function and behavior are discussed. Finally, the promising therapeutic potential of psychobiotics to modify gut microbial ecology in order to improve mental well-being is presented. Here, we show that the individual contribution of nutrients, their interaction with other micro- and macronutrients and the way in which they are organized in the food matrix are of crucial importance for normal neurocognitive development.

## 1. Introduction

Nutrition is critical in supporting healthy brain development early in life, with long-lasting, and often, irreversible effects on an individual’s cognitive development and life-long mental health. In this review, we present recent human and pre-clinical evidence on the role of nutrition, with particular focus on more emerging nutrients, in neurocognitive development in healthy infants and children aged 0–59 months. 

Development of the human brain starts with the closure of the neural tube by the fourth week of pregnancy [1] and the proliferation of neurons in the germinal layers near the ventricles during the early phases of gestation (from week six of pregnancy) [2]. This is followed by the migration of neurons to their final destination and simultaneous initiation of neuronal differentiation (Figure 1, adapted from [3]). Neuronal differentiation includes the formation of dendrites and axons, the production of neurotransmitters, the development of synapses and intracellular signaling systems, and establishment of complex neural membranes starting from late pregnancy until the first few months postnatally. The formation of synapses continues throughout life [2,4], whereas the production of various neurotransmitters starts prenatally and reaches mature levels around the age of three years [5]. In parallel, glial cell production begins during the second trimester (32 weeks of gestation) [2,5]; glial cells insulate the axons by surrounding them with a membranous myelin sheath (axonal myelination), a process that predominantly takes place between the second trimester of gestation and the first year of life. Myelination continues, but starts to decline in early adulthood after which it stops at around the age of 40 years [2]. 

Development of brain structures occurs in several phases. The major transition takes place when the transient cortical structure, mediating fetal and neonatal behavior, is replaced by the cortical plate at three to four months of age. As a result, motor behavior changes from non-directed general movements to more goal-directed movements, such as reaching. The hippocampus, which is important for facial and scene-recognition, as well as spatial memory, develops at approximately 32 weeks of gestation until at least 18 months postnatally [4,5]. The prefrontal cortex, responsible for complex processing tasks such as attention and multi-tasking, exhibits initial rapid development during the first 6 months of life [5]. Note, that prefrontal development continues well into the third decade of life [6]. The pruning of axons and synapses to further optimize the brain’s functioning usually starts between the onset of puberty and early adulthood [2].

The brain’s structure and capacity is formed relatively early in life, before the age of three years [5]. A failure to optimize brain development at this important stage of life can therefore have negative long-term consequences in terms of education, work potential, and mental health (the “ultimate cost to society” of early life adversity).

## 2. Nutrients that Play a Role in Neurodevelopment

### 2.1. Lipids

#### 2.1.1. Long-Chain Polyunsaturated Fatty Acids

Neurodevelopment is influenced by a number of factors ranging from gestational age at birth and social environment to nutrition. Dietary fat in particular is an important modifiable nutritional factor illustrated by the crucial role of the polyunsaturated fatty acids (PUFAs) linoleic acid (LA; 18:2 n-6), α-linoleic acid (ALA, 18:3 n-3), docosahexanoic acid (DHA; 22:6 n-3) and arachidonic acid (AA, 20:4 n-6) in normal brain formation and neuronal myelination during infant neurodevelopment [2,7]. Furthermore, long-chain PUFAs (LCPUFAs) have been shown to affect the production of various neurotransmitters [5], with profound effects on monoaminergic, cholinergic, and gamma-aminobutyric acid (GABA) ergic systems. DHA is especially important for visual and prefrontal cortex development, the latter of which mediates attention, inhibition and impulsivity actions [5]. 

In addition to sufficient intake via diet or supplementation, a balanced ratio between LA and ALA is important as well [8]. In the prospective Mothers and Children’s Environmental Health (MOCEH) cohort of 960 pregnant women in Korea, an inverse association between LA/ALA ratio during pregnancy and Mental Developmental Index (MDI) and Psychomotor Developmental Index (PDI) scores in the offspring at six months of age was reported [7]. The average LA/ALA ratio in this population amounted to 11.12 ± 6.9. 

LA is abundantly present in daily food and high intake can have negative health consequences [8]. High LA in colostrum and breastmilk has been associated with poorer motor and cognitive scores at two and three years of age [9] and a lower verbal intelligence quotient (IQ) at five-to-six years [10] in the observational EDEN (Etude de cohorte généraliste, menée en France sur les Déterminants pré et post natals précoces du développement psychomoteur et de la santé de l’Enfant) cohort in France. This negative impact was postulated to take effect through several mechanisms, namely the suppression of biosynthesis of n-3 PUFAs (due to enzymatic competition to convert n-6 and n-3 PUFA to LCPUFA), which supplies necessary DHA for brain development, as well as by decreased uptake of circulating DHA by the brain and thus impaired accretion of DHA in the brain. Lastly, n-6 PUFAs are precursors for several pro-inflammatory eicosanoids that can be produced in early life and may have a negative impact on cognitive function [10].

Several observational studies reported that high DHA levels in pre- and postnatal periods seem to improve specific cognitive skills ranging from processing ability and attention to overall IQ in the offspring, even up to 12 years of life [11,12,13]. Nonetheless, the impact of DHA supplementation during pregnancy remains controversial. Daily 400 mg DHA supplementation for 20 weeks in pregnant mothers showed positive effects on the infant’s attention ability at 5 years of age [14], while an earlier study among 2499 pregnant women in Australia found that daily 800 mg DHA supplementation did not affect cognitive and language development of the offspring at 12 and 18 months [15]. The latter is further supported by the results of a Cochrane systematic review (2015) stating that there was no effect of DHA supplementation in breastfeeding mothers on language development, problem-solving abilities, psychomotor development, and general movement ability of their offspring [16]. Another Cochrane review (2017) reported that, although there was no concern over the safety of infant formulas supplemented with DHA and AA, the majority of evaluated randomized controlled trials (RCTs) did not show any beneficial effects on neurodevelopmental outcomes in term infants [17]. In addition, the authors concluded that the positive effect on visual acuity had not been consistently demonstrated. Although the review included the study by Colombo et al. [18], reporting a beneficial effect of DHA/AA supplementation (up to 0.64% DHA/0.64% AA) in infant formula on problem-solving skills at nine months [18], it did not include earlier studies on the topic [19,20]. These studies indicated that routine supplementation of term infant milk formula with DHA (at the level of 0.3% PUFA), from birth to four months of age, was associated with improved neurodevelopmental outcome at four months [20] and higher MDI scores at 18 months [19]. In addition, better inhibitory control measured by behavioral and brain electrophysiology responses among those supplemented with the above-mentioned dose at 5.5 years of age has been described as well [21]. Importantly, dosing LCPUFAs at a level higher than 0.64% in early life may have negative effects on cognitive development at a later stage [18].

The impact of essential fatty acids (EFA) and LCPUFAs on cognition and brain development appears to be particularly evident in older children. Interestingly, DHA and AA supplementation of 200 mg daily in Growing Up Milk (GUM) among toddlers aged 13 months for the duration of one year increased the Bayley III language composite score at 24 months as compared to those receiving standard GUM without LCPUFA. The same study reported fewer inattention episodes among boys receiving LCPUFA-supplemented GUM as compared to their unsupplemented counterparts [22]. 

Two separate studies in two-to-six years old children in Ghana and Tanzania revealed that children with the highest levels of blood EFA and DHA had at least a three times higher chance of successfully passing an executive function test [23,24]. In older pre-school children, the consumption of 978 g of fish over one week influenced cognitive function compared to those consuming 850 mg of meat, after adjusting for dietary compliance; information on EFA and DHA blood levels was not included [25]. 

#### 2.1.2. Polar Lipids

Polar lipids are amphiphilic in nature and contain a hydrophobic tail and a hydrophilic head. Phospholipids (glycerophospholipids and sphingomyelin) and sphingolipids (ceramides, cerebrosides and gangliosides) are the main representatives of this group. Polar lipids make up biological membranes but are also found in circulating fluids. In mammalian milk, the milk fat globule membrane (MFGM), the trilayer membrane structure surrounding each fat globule, is an important source of polar lipids [26,27], as are nanovesicles (exosomes). Nanovesicles are secreted into milk by mammary gland cells and are implicated in cell-to-cell communication by virtue of their functionally active cargo (e.g., messenger-RNA (mRNA), micro-RNA (miRNA), and different proteins; [28]). Human as well as bovine milk contains approximately 4% fat in the form of fat globules [29]. These globules are filled with triglycerides, constituting 98% of total fat. The lipids within the MFGM primarily include polar lipids but also comprise neutral lipids (like cholesterol). The main polar lipids present in human and bovine MFGMs are phosphatidylcholine (PC), phosphatidylethanolamine (PE), phosphatidylinositol (PI), phosphatidylserine (PS), and sphingomyelin (SM) [30,31]; human milk contains higher levels of SM and PS, whereas relatively more PE is present in bovine milk fat [32].

To date, no intake recommendations or guidelines for polar lipids have been proposed or implemented by health authorities. However, adequate intakes have been defined for two nutrients that serve as structural parts of polar lipids, choline and DHA [33,34]. Presently, only limited scientific evidence exists on the brain bioavailability of polar lipids via placental transfer or transport over the blood brain barrier (BBB) [35,36,37]. Nevertheless, supplementing polar lipids in wild-type animal models and healthy infants does suggest benefits for cognitive performance.

The putative role of MFGM polar lipids in brain and neurocognitive development has received significant attention. In Sprague-Dawley rats, oral gavage supplementation with MFGM led to neurocognitive benefits by early upregulation of genes involved in brain function, such as brain-derived neurotrophic factor (BDNF) and St8 alpha-N-acetyl-neuraminide alpha-2,8-sialyltransferase 4 [38]. Somewhat unexpectedly, a human RCT evaluating the effects of maternal dietary supplementation of complex milk lipids (CML; gangliosides and phospholipids) from the MFGM during pregnancy on fetal growth showed no effects on any of the fetal biometric dimensions measured [39]. The lack of effect could be due to the application of an inadequate dose of polar lipids. In a Belgian study, a phospholipid-rich MFGM concentrate given daily to preschool children aged 2.5–6 years for a period of four months decreased behavioral problems and reduced days with fever during the intervention period [40]. Notwithstanding this encouraging result, Timby and colleagues (2017) concluded that while MFGM interventions seem safe, it is still unclear which MFGM fractions are most suitable for supplementation and at what concentration at which age. Furthermore, it was pointed out that the evidence base for the effects of MFGM polar lipids on brain and neurocognitive development is still limited [41]. Since then, several studies have provided additional evidence for the importance of the MFGM in early life with mixed outcomes. A study in 451 healthy term infants showed that receiving formula with added bovine MFGM and bovine lactoferrin (LF) resulted in accelerated neurodevelopment at day 365 as evidenced by higher mean cognitive (+8.7), language (+12.3), and motor (+12.6) Bayley-III scores, and improved global development scores from day 120 to day 275 and attention at day 365 in the MFGM + LF group [42]. In addition, enhanced language skills at day 545 were observed (some subcategories of the MacArthur-Bates Communicative Development Inventories were higher in the MFGM + LF group). 

There has been growing interest in the use of gangliosides as part of a supplement for either the infant or the mother because these polar lipids serve a crucial role in pre- and postnatal development of the brain, which coincides with the critical window of rapid brain growth around birth [43]. Ceramides are essential for neural development contributing to ganglioside synthesis in utero. Interestingly, the phlorizin domain of the lactase enzyme splits ceramides from glucosyl, galactosyl, and lactosyl-cerebrosides [44]. In addition, lactase splits lactose into glucose and galactose. As nearly all healthy infants are lactase persistent, lactase activity in their small intestine yields ceramide, glucose, and galactose moieties from dietary lactose and glycosphingolipid intake, all important building blocks for the developing nervous system. Gangliosides are glycolipids that contain sialic acid, which is an essential nutrient for optimal brain development and cognition. Endogenous production of sialic acid is possible but limited. Rather, it is available in human milk oligosaccharides in relatively large quantities, predominantly in the form of Neu5Ac (N-acetylneuraminic acid), which is the precursor of various neural brain glycoproteins, including polysialic acid, gangliosides, glycosaminoglycans, and mucins. The major protein carrier of polysialic acids is NCAM (Neural Cell Adhesion Molecule); polysialylated-NCAM is a key neuroplastic molecule involved in neuronal plasticity and of crucial importance for memory formation. Other examples of sialylated proteins are synaptic cell adhesion molecule 1 and scavenger receptor CD36 [45]. Sialic acid is particularly abundant in neuronal cell membranes. Notably, the location and amount of sialic acid in different regions of the brain change dramatically during development. 

The high sialic acid content of human breast milk in addition to its role in brain development suggest that sialic acid in breast milk has an impact on infant cognitive development. This could also imply that brain growth creates a greater need for sialic acid than can be provided by endogenous biosynthesis in the infant. This is supported by findings from a study in which sialic acid was measured in brain samples from infants (1–38 weeks) that died of sudden infant death showing that the sialic acid content was higher in the brains of breast-fed infants than in those of formula-fed infants [46].

### 2.2. Minerals

#### 2.2.1. Iron

In addition to lipids, micronutrients are critical for normal neurodevelopment as well. Iron, for instance, is vital for energy production, oxygen transportation, and DNA synthesis. It plays a crucial role in hippocampal development, myelination and production of neurotransmitters, such as dopamine, serotonin, and norepinephrine as shown in pre-clinical studies [5,47,48]. Iron deficiency results in reduced 6-desaturase enzyme activity that is required for the synthesis of essential fatty acids and can therefore impair the synthesis of α-linoleic acid (ALA, 18:3 n-3) into DHA [49]. Currently, no literature is available on the impact of combined iron and LCPUFA deficiencies on cognitive neurodevelopment.

Limited evidence from recent studies suggest that iron supplementation during pregnancy and infancy may positively influence the psychomotor development of children [47,48,50]. A positive effect of prenatal iron supplementation during pregnancy on overall cognitive development of the child has been described only for anemic pregnant women. In this group, a favorable effect on cognitive performance in children under two years of age, toddlers and primary school children was observed [48,50,51].

Based on a systematic review and a follow-up study, the effects of iron supplementation during early life on cognitive function are unclear at 12 months of age. In addition, no benefit on cognitive function at 18 months could be detected [47,50]. Notably, the provision of iron to iron-replete infants could have a negative effect on long-term cognitive development as shown in a cohort study among infants in Chile [47,52]. Positive impact of iron supplementation on cognitive function seems to be observed only in anemic primary school children [47,51]. Based on the available evidence, adequate dietary iron intake should be encouraged during pregnancy and post-natal life up to adulthood. With regard to iron supplementation, a different picture emerges. Given the uncertainty of the efficacy of iron supplementation due to significant supplementation heterogeneity across the various studies (i.e., type and format of iron supplementation, dosage, length of supplementation, presence of other nutrients) [47,50], combined with the potential negative impact of providing high iron dosages to iron-replete infants [52], it seems advisable to restrict the provision of iron supplements, in the right form and dosage, to anemic individuals.

#### 2.2.2. Zinc

Zinc is necessary for central nervous system (CNS) development and is one of the most ubiquitous metals found in the brain; it is present in many enzymes involved in brain growth and is important for neurotransmission. In animal models, zinc has been shown to be involved in neurogenesis, neural migration, synaptogenesis, and regulation of GABA-Ergic neurotransmitter release [5,53].

Zinc deficiency during pregnancy and early infancy has long been associated with developmental deficits, such as poorer learning, attention, memory, and mood [5]. However, there is currently no convincing evidence that maternal zinc supplementation improves cognitive development in the offspring [54,55].

Evidence on the association between zinc intake and cognitive development is quite limited for infants. Six-month old infants receiving either a combination of micronutrients (10 mg/d zinc, 10 mg/d iron and 0.5 mg/d copper) or iron and copper alone for the duration of 6 months presented with different outcomes. In the group receiving zinc, there was an improvement in normative information processing and active attentional profiles at two years of age. However, no differences were reported regarding other parameters, which included Bayley Scales of Infant Development (BSID) at six, 12, and 18 months. Infants receiving zinc supplementation were also able to maintain a better zinc status [55].

A Cochrane systematic review (2012) concluded that there is no significant effect of zinc intake on mental and motor development in children [56]. This was nuanced by another systematic review, published later that year, stating that the effect of zinc supplementation on cognitive function might be dependent on the dose of supplementation and the duration of the intervention [57].

#### 2.2.3. Iodine

Iodine plays an important role in brain development in the form of thyroxine and triiodothyronine. It affects the timing of differentiation of neural tissue in the brain prenatally and determines the number of glial cells for myelin sheath production postnatally. Maternal thyroid hormone can be found in the embryonic cavities at the end of week 4 post-conception when the formation of the brain cortex and the anterior part of the neural tube takes place [58]. 

The impact of pre-conception iodine levels has been recently investigated in the Southampton Women Cohort [59]. The results revealed that a low maternal urinary iodine concentration, measured by iodine/creatinine (I/Cr) ratio, at 3.3 years before conception was associated with low overall childhood cognitive function at 6–7 years as assessed by the Wechsler Abbreviated Scale of Intelligence (WASI) [59]. Around 8.9% of the women in this cohort presented with a low I/Cr ratio. Unexpectedly, the same study reported no influence of pre-conception iodine levels on specific measures of executive function at the age of six-to-seven years [59].

The Generation R cohort study showed that mild-to-moderate iodine deficiency in early pregnancy affects the offspring’s behavior and risk for development of ADHD at eight years of age [60]. Severe iodine deficiency during pregnancy is well-known to result in maternal and fetal hypothyroidism and has been shown to be associated with serious adverse health effects in the offspring, including congenital hypothyroidism, growth retardation and impaired cognition encompassing deficits in hearing, speech, gait and IQ [5,61]. Still, several iodine supplementation studies during pregnancy on offspring cognitive function reported inconclusive findings. This, in part, may be explained by the application of age-inappropriate global development assessments that may have caused misclassification and lack of correlation with cognitive function at that particular time [58,62,63].

Post-natally, iodine continues to play a role in neurocognitive development. The level of iodine in colostrum predicts the motor development capability of infants at 18 months, but does not relate to other abilities, such as language development or overall cognition [64]. Interestingly, a study on iodine supplementation using iodized salts for children in areas where the incidence of iodine deficiency is high, reported no benefits on cognitive function in children older than three years of age despite the improvement in iodine status [65]. 

### 2.3. Vitamins

Despite extensive research conducted on vitamin supplementation, only limited recent evidence exists to suggest that vitamin supplementation during pregnancy and early intake by infants positively influences cognitive development of children. Nevertheless, vitamin A, vitamin B12 (cobalamin), folate, and vitamin D are well-recognized for their capacity to critically influence early cognitive development [5,49], and micronutrient deficiencies early in life can lead to impairments of the CNS. 

#### 2.3.1. Vitamin A

Mice postnatally deprived of vitamin A exhibited a reduction in the expression of brain retinoid receptors and associated target genes, which was accompanied by selective memory impairment after 39 weeks of deprivation [66]. Retinoids have typically been associated with relational memory, synaptic plasticity, learning, memory and sleep, and are essential nutrients that help support normal embryonic development, cell growth, and differentiation [49].

A recent human intervention study on vitamin A and cognitive function assessed the effectiveness of vitamin A, zinc, glutamine, zinc plus glutamine, zinc plus vitamin A, and vitamin A plus zinc plus glutamine [67]. Only the combination of all three positively influenced cognitive function in girls (aged 6–12 years), but not in boys. It should be noted, however, that the power of this study was limited due to the small sample size of each intervention group.

#### 2.3.2. Vitamin B12

A study on maternal intake of methyl-donor nutrients and child cognition at three years of age revealed a weak inverse association for vitamin B12 intake and a linear association for folate intake during the first and second trimester with the Peabody Picture Vocabulary Test III (PPVT-III scores) [68]. Each 600 mcg/day increment in total folate intake during the first trimester was associated with an increase of 1.6 points in PPVT-III scores. No correlations were found between choline, betaine, or methionine and cognitive function [68]. 

Recent findings from the GUSTO (Growing Up in Singapore Towards healthy Outcomes) cohort showed that maternal B12 deficiency was associated with lower cognitive scores of infants at 24 months when compared to infants from vitamin B12-replete mothers [69]. The level of vitamin B12 at 2–12 months correlated with development and performance in social perception tasks and visuo-spatial abilities at 5 years of age among 330 children in Nepal [70]; an increase of one unit in vitamin B12 status was associated with an increase of 4.88 in the Ages and Stages Questionnaires (ASQ-3), 0.82 in recognition score, 0.59 in geometric puzzle score, and 0.24 in block construction scores. In another study (650 children, India), vitamin B12 status at four-months was associated with increased BSID-II score at 12–14 months [71]. Vitamin B12 plays an important mechanistic role in neural myelination, synaptogenesis, and neurotransmitter synthesis in pre- and postnatal periods. In addition, it promotes development of the hippocampus and is therefore relevant to memory, language, and visual processing [69]. Vitamin B12 and folate are required for cell division and generation of methionine, which is needed to produce neurotransmitters and myelin [70]. A relationship between folate and vitamin B12 was also reported by Strand and colleagues, demonstrating that the plasma folate concentration was independently associated with mental development scores when children with poor vitamin B12 status were excluded from the analysis [72]. Timing of folic acid intake (and the resultant endogenous folate level) is likely critical, as it is well-known that severe folate deficiency during pre-conception and early pregnancy is associated with inadequate closure of the neural tube resulting in severe brain defects, including spina bifida [1]. However, in older children, supplementation with folic acid, vitamin B2, B5, and calcium did not affect verbal IQ, short term memory, or processing speed [73]. 

#### 2.3.3. Vitamin D

In a rodent model, low maternal vitamin D (1,25-dihydroxycholecalciferol) status during pregnancy was associated with structural changes in the brain, such as enlarged lateral ventricles, a thinner cortex, and increased cell proliferation [49,74]. Low prenatal vitamin D status was also linked to the severity of schizophrenia and autism symptoms in epidemiological studies [75,76]. In addition, vitamin D status during pregnancy was shown to be related to cognitive development, and maternal 25(OH)D levels <50 nmol/L were independently associated with low MDI and PDI scores at six months [77]. This prospective cohort of 363 mother-infant pairs in China also reported an inverted U–shaped relation between vitamin D levels in cord blood and neurocognitive score at 16–18 months [77]. Interestingly, vitamin D has been shown to be able to upregulate serotonin expression [78].

### 2.4. Dietary Protein and Amino Acids

A classic example of the importance of proteins to behavioral and neurocognitive development in infants is the longitudinal study by Chavez et al. [79] which evaluated the effects of nutritional supplementation on infants’ physical, mental, and social development in two groups of 17 mother-child pairs in a poor rural Mexican community. In this study, one group of mothers was supplemented daily with 205 calories and 15 g of protein during pregnancy and 305 calories and 15 g of protein during lactation, whilst the other group followed the usual feeding habits of the community. Between the 12th and 16th week of life, the supplemented infants began to receive whole cow’s milk *ad libitum* and prepared baby food in quantities sufficient to maintain adequate rates of growth. At 18 months of intervention, the mothers of supplemented children displayed more complex interactions with their children, who were more restless, playful, demanding and disobedient than those non-supplemented. These results suggest a beneficial effect of protein (and energy) supplementation on the behavioral patterns within the family, with the more active children eliciting greater stimulation from their parents. Another historic study [80] demonstrated that protein supplementation, rather than energy, during early childhood improved psycho-educational performance. Therefore, Guatemalan children exposed to protein supplement scored significantly higher on tests of knowledge, numerical aptitude, reading and vocabulary as compared to those that only received energy supplementation. Two decades later, 130 female subjects were re-evaluated and, interestingly, women exposed to protein supplementation during early childhood had better educational achievements than those from the energy group [81]. 

#### 2.4.1. The Importance of Protein Quality

The nutrient value of dietary proteins (protein quality) essentially resides in the individual (essential) amino acids that are absorbed into the system. During gestation, the growing fetus only receives amino acids because proteins, with the exception of certain immunoglobulins, do not cross the placenta in significant amounts. Amino acids are the precursors of structural proteins required for the growth of body tissues, including the brain. In addition, various amino acids are precursors of neurotransmitters or, in many cases, are neurotransmitters themselves. They also serve as direct precursors of enzymes and peptide hormones. Therefore, insufficient provision of any single amino acid from the maternal diet can hinder protein synthesis by the fetus and can have deleterious effects on fetal brain development similar to those induced by the omission of proteins as a whole [82]. Rodent studies studying inadequate protein intake (protein malnutrition) report reduced brain size, dendritic arborization, cell maturation [83], emotional reactivity and sensitivity to aversive or painful stimuli [84], reduction in cognitive flexibility [83,84], and learning and memory impairments [85,86,87].

#### 2.4.2. mTORC1: Linking Protein (in)Adequacy to Brain Development

One plausible mechanistic pathway that may explain the link between protein (in)adequacy and brain development is the activity of the serine/threonine kinase mammalian target of rapamycin complex 1 (mTORC1), which is a master regulator of all cell growth and metabolism. This kinase integrates signals triggered by different stimuli, such as variations in amino acid supply, changes in the cellular energy state, growth factors (e.g., BDNF, insulin, and IGF1 (insulin-like growth factor 1)), and (within the brain) by transduction of neurotransmitters and neurotrophin signals [88,89,90]. After influx through L-amino acid transporters, leucine activates mTORC1 in neurons [91]. In addition, uptake of arginine by the cationic amino acid transporters CAT1 and CAT3 has also been demonstrated to activate mTORC1 in neurons [92]. Amongst growth factors, insulin and IGF1 enhance mRNA translation in neurons possibly through mTORC1 [93,94,95,96]. BDNF, the most prominent neurotrophic factor in the CNS [97,98], has been shown to activate mTORC1 signaling and enhance *de novo* protein synthesis in cortical neurons [99,100]. Several studies suggest that amino acid sufficiency is essential for the insulin-induced activation of mTORC1 in several cell lines [101,102], but not for BDNF-induced mTORC1 activation in neurons [103]. Neurotransmitters such as serotonin (5-HT) have also been reported to possibly activate mTORC1 [104,105,106]. During early CNS development, mTORC1 is involved in neural stem cell proliferation, migration, and differentiation, axonal and dendrite development, gliogenesis, synaptic plasticity, and learning and memory storage [107,108]. Aberrant mTORC1 signaling alters neural development and can result in a wide spectrum of neurological developmental disorders, including learning disabilities and mental retardation.

#### 2.4.3. Tryptophan

Recent studies suggest that essential amino acids like tyrosine and tryptophan could be important factors in neurodevelopment. Tryptophan is the substrate for 5-HT (serotonin), which has been implicated in the control of mucosal secretion and absorption of nutrients and acts as a vasodilator and regulator of motor and sensory functions, including perception of pain and nausea [109,110]. In the CNS, 5-HT is involved in the regulation of mood, behavior, and cognitive functions [111,112]. Anomalies in serotoninergic transmission have been linked to psychiatric disorders, such as major depression and schizophrenia [113]. Further metabolic reactions downstream of 5-HT yield melatonin [110,114], a hormone known to regulate circadian rhythms of behavior, physiology and sleep patterns [110,115]; all events essential for proper development of the brain. There is some limited evidence to suggest that feeding infant formula enriched with tryptophan improves sleep, which in turn may influence neurobehavioral development [109,116]. In an early study, 20 healthy newborns (aged 2–3 days) were randomly assigned to receive either formula or a glucose solution containing tryptophan or valine. Infants who were fed the tryptophan solution entered active sleep sooner than those receiving formula, while infants receiving the valine solution entered sleep much later [117]. Oral administration of tryptophan to infants also has been shown to increase urinary levels of serotonin and melatonin [118], and, similar to findings in rats, administration of tryptophan at night is known to increase circulating concentrations of both serotonin and melatonin [119]. In a subjective assessment of 1055 infants and children aged 0–6 years in Japan, Harada et al., (2007) reported a correlation between tryptophan intake (at breakfast) and mood/behavior as well as sleep patterns [120]. Metabolism of tryptophan to 5HT during the daytime and further conversion to melatonin at night was proposed as a possible mechanism for the observations. 

Approximately 3% of daily tryptophan is metabolized into 5-HT and about 90% is processed along the kynurenine pathway rendering molecules often collectively referred to as ‘kynurenines’, while the remainder is degraded by the gut microbiota to produce indole and its derivatives [121]. Kynurenines have been gaining interest because of their role in the developing brain [122]. The kynurenine pathway regulates the production of neuroprotective (e.g., kynurenic and picolinic acid, and the essential cofactor NAD+) and neurotoxic (e.g., quinolinic acid, 3-hydroxykynurenine) metabolites [122,123], and is controlled by two key rate-limiting enzymes: TRP-2,3-dioxygenase and indoleamine-2,3-dioxygenase (IDO-1) [124]. The activity of IDO is regulated by proinflammatory cytokines, such as interferon-γ, released in response to toll-like receptor (TLR) activation [125], suggesting that the kynurenine pathway is more active during periods of immune activation or under pathological conditions [126]. An increase in tryptophan metabolism via the kynurenine pathway reduces the availability of tryptophan for 5-HT synthesis and increases production of harmful kynurenine metabolites in the brain, contributing to mood disorders [127]. In addition, it may result in decreased melatonin levels, which are associated with circadian malfunctioning and can increase the risk of mood disorders as well [128]. These links between kynurenine pathway metabolism and brain development are particularly relevant when considering children in low- and middle-income countries (LMICs) with higher rates of infection and/or chance of exposure to a range of other harmful environmental factors [126,129,130,131]. As accumulating evidence indicates that changes in kynurenine metabolism affect brain development [132,133], with consequences that persist into adulthood [132,134,135], it can be hypothesized that any infection during the very early stages of brain development may result in CNS dysfunction, or increased susceptibility to dysfunction, later in life.

#### 2.4.4. Tyrosine and Phenylalanine

Phenylethylamine (PEA), a neurotransmitter and hormone, is metabolized from phenylalanine and increases extracellular levels of dopamine (DA), modulates noradrenergic transmission, and may act as a neuromodulator for catecholamines [136]. The preferred substrate for the synthesis of catecholamines (DA, norepinephrine (NE), and epinephrine) is tyrosine which is synthesized by hydroxylation of phenylalanine, and thus, not considered essential. However, if the hydroxylase system is deficient or absent, tyrosine requirement must be met by the diet [137]. The rates of synthesis and release of catecholamines directly rely on the brain concentration of tyrosine, which in turn is affected by tyrosine availability from blood [137]. Uptake from the blood into the brain takes place through a common carrier that tyrosine shares with a number of other large neutral amino acids (LNAA), including tryptophan, phenylalanine, and the branched-chain amino acids (BCAAs; leucine, isoleucine, and valine), as well as ingested protein (not just from a single meal but over several days) [138].

Some evidence on the functional effects of rapid reductions in brain tyrosine levels exists. Consistent with the reduced DA release due to tyrosine depletion, reports suggest plasma prolactin concentrations are increased [139], features of spatial recognition memory and performance are impaired [140], and the subjective psychostimulant effects of amphetamine are attenuated [140]. Tyrosine depletion has also been reported to lower some indices of mood (and promote apathy) in normal subjects [141]. It is, however, important to note that all studies demonstrating functional effects of tyrosine-related changes in catecholamine production have been pharmacological in nature; physiological relevance has yet to be demonstrated and validated. One study in two to five-year-old children suggested that ingestion of >800 mg tyrosine or phenylalanine at breakfast tended to lower the frequency of depressive state [142]. 

#### 2.4.5. Branched Chain Amino Acids

The BCAAs leucine, isoleucine, and valine also participate directly and/or indirectly in a variety of important biochemical functions in the brain, such as protein synthesis, energy production, compartmentalization of glutamate (an excitatory amino acid neurotransmitter in the brain), and synthesis of the amine neurotransmitters 5HT, DA, and NE, which are derived from the aromatic amino acids [143]. The role of leucine as a trigger for mTORC1 activity was described earlier. However, the available evidence linking dietary BCAA intake to brain function is limited to the production of the amine neurotransmitters [143]. This may be because a major fraction of BCAAs ingested during a meal passes into the systemic circulation causing plasma concentrations to rise considerably, increasing their uptake into the brain and decreasing the uptake, as well as level of the aromatic amino acids (see above). Consequently, the synthesis and release of 5HT and catecholamines is reduced [144,145]. The functional effects of such neurochemical changes include altered hormonal function, blood pressure, and affective state. Although BCAAs appear to have clear biochemical and functional effects in the brain, few attempts have been made to characterize time or dose-dependent relations for such effects. To the best of our knowledge, no studies have been conducted to identify levels of BCAA intake that might produce adverse effects on the brain in a normal pediatric population.

## 3. Nutrient Interactions through the Gut-Brain Axis (GBA)

### 3.1. What Is the Microbiome Gut-Brain Axis?

Approximately 2500 years ago, Hippocrates stated that all disease begins in the gut. This bold statement has now received support from a substantial body of research in both humans and animal models. The gut is home to a complex and dynamic ecosystem consisting of trillions of microorganisms—the gut microbiota, including bacteria, archaea, yeasts, viruses and protozoa [146]. The human host and the gut microbiota are conceptualized as a unique entity [147]. Whereas humans shape the microbiota via diet and life-style changes [148], the microorganisms reciprocally contribute to the host physiology. This symbiotic relationship starts from the moment when the maternal microbiota shape embryonic development [147] and initiate infant gut colonization during birth, and is continued by the role of the infant microbiota in postnatal brain and cognitive development [149,150,151], as well as its important, longer-term influence on the maturation of immune, endocrine and neural systems [152,153,154].

The gut and the brain are intimately connected via the gut-brain axis (GBA), which involves several bidirectional communicational routes. Firstly, the gut presents with a wide-extended enteric nervous system (ENS). The gut bacteria can modulate ENS electrophysiological thresholds [155], as well as influence ENS development via the activation of pattern recognition receptors (PRRs) [156]. Accordingly, germ free (GF) mice, which are devoid of microorganisms and can, thus, show the effects of gut microbiota on host physiology [157] display ENS abnormalities in the postnatal period [158]. Beyond influencing the nervous system via the ENS, the microbiota communicate with the brain via autonomic, immune, endocrine and metabolic pathways [159,160,161]. Specifically, gut microbiota have been suggested to regulate gene expression in the brain by inhibiting miRNA and mRNA translation [162]. Research has also shown that the increased or decreased expression levels of many miRNAs could reflect the various pathophysiological processes of diseases by way of neurotransmitter expression, via the synthesis and release of neurotransmitters implicated in psychopathology (e.g., GABA and the precursor pool for serotonin [163], as well as BDNF expression [164]). These neuroactive metabolites (e.g., GABA) can directly interact with gut autonomic synapses [165], and thereby, modulate brain neurochemistry and behavior. In addition, microbes also act via the production of short-chain fatty acids (SCFAs) and peptidoglycan (PGN). SCFAs include acetate, butyrate, and propionate and can have wide-range effects on host health, from gastro-intestinal functioning to body metabolism. Indeed, while 90–95% of SCFAs are promptly absorbed after production and utilized by either the gut mucosa or the liver, a minor fraction –mostly acetate reaches the systemic circulation [166,167] by which they can exert various effects on the brain. This is supported, for example, by the observation that butyrate injections increase central serotonin and BDNF in mice [168] and by the presence of acetate, propionate and butyrate metabolites, in decreasing concentrations, respectively, in human cerebrospinal fluid. Therefore, SCFA signaling occurring in response to microbiota changes in the intestine can reach distant organs, including the brain, and can influence the inflammatory and metabolic state [169]. Further, PGN can cross the blood-brain barrier (BBB) and interact with PGN-sensing molecules and transporters. It has been suggested that the influence of gut microbiota might even start prenatally by contributing to BBB integrity regulation [170]. Noticeably, a study in healthy animals reported increased PGN concentrations in the brain in parallel with postnatal bacterial colonization, providing evidence for the role of microbiota in early neurodevelopmental processes [171]. Interestingly, these age-dependent increases in PGN concentrations were found across different brain regions, such as the prefrontal cortex, the striatum and the cerebellum, which suggests that these mechanisms are domain-general and affect the entire developing brain.

### 3.2. The Influence of the GBA on Cognitive, Emotional and Neural Development 

Evidence from animal models has shown that the gut microbiome not only influences initial brain development, including synaptogenesis and myelination of brain areas supporting motor functions and cognitive abilities [150], but that it also affects brain responsiveness and function across the entire life-span by regulating neurotransmitter, synaptic, and neurotrophic signaling systems and neurogenesis [157]. While infants are initially born with sterile intestines, bacterial colonization begins with the process of birth and continues to proceed rapidly [172], with significant increases in both diversity and functional capacity over the first few years of life [173]. In addition, the gut microbial composition varies across the age span, with the developing microbiome being characterized by a relative abundance of genes that support functional and structural brain development. Whereas, the adult microbiome shows a predominance of genes related to inflammation, obesity or adiposity [174]. Interestingly, recent investigations are now also highlighting the important role that social networks, such as immediate family or social peers, play in microbial transmission and development [175]. Over the last decade, much research on the gut microbiome in humans has therefore focused on characterizing microbe populations in both health and disease to identify possible avenues for intervention [146,159,176,177,178].

With regards to cognitive functioning and mental health, it has been shown that a significant alteration of microbial diversity, known as ‘dysbiosis’, affects brain-behavior relationships and may result in behavioral and cognitive abnormalities. This is particularly critical in development, where ongoing maturation and increased plasticity levels may lead to atypical behavioral patterns and abnormal brain network maturation [179,180,181]. Indeed, the first evidence from animal models provides support for the importance of this critical period of microbiome GBA development. For example, research has highlighted the significant impact of microbiome dysbiosis on brain networks that support the regulation of emotion during development [176,182,183]. In addition, infant rats that had been given phytohemagglutinin or microbial protease did not only show precocious gut barrier maturation but also gut dysbiosis, which seemed to favor low grade systemic inflammation and, ultimately, affect BBB maturation, possibly via changes in SCFA concentrations [184]. Further, GF mice exhibited lower levels of mRNA expression in the prefrontal cortex (PFC), amygdala and cingulate cortex compared to specific pathogen free mice [176]. These findings were complemented by the results of another study, demonstrating that the development and function of the amygdala is altered by the microbiota, as gene expression within the amygdala presented a different developmental trajectory in GF mice versus controls [183]. Importantly, early deficits in gut microbiota may not be reversible as indicated by the observation that colonization of gut bacteria in adolescent GF mice was not sufficient to reverse anxiety-like behavior [182]. These findings further highlight the relevance of understanding the changes in the GBA during early development.

To date, only a few studies have looked at the role of the gut microbiome in the development of cognitive and psychological functioning in infants and children. In one study, the association between the gut microbiota composition (GMC) and temperament of 2.5-month-old infants was evaluated [185]. Infants with a *Bifidobacterium*/*Enterobacteriaceae* GMC exhibited higher regulation scores, whereas a *Streptococcus*/*Bifidobacterium* GMC was associated with positive emotionality. In addition, infants with reduced microbial diversity presented with higher negative and fear reactivity. Interestingly, the detected GMC-temperament associations appeared to be sex-dependent. These results were in line with an earlier study demonstrating that microbial composition at the age of 18–27 months is differentially associated with infant temperament in girls and boys [186]. More specifically, higher levels of phylogenetic diversity in the gut microbiome were correlated with higher levels of surgency/extraversion on the infant temperament scale in both genders. Whereas, only boys showed an association with differences in beta diversity, the Shannon Diversity Index (SDI), and the abundances of *Dialister*, *Rikenellaceae*, *Ruminococcaceae*, and *Parabacteroides* [186]. In another study, a longitudinal design was implemented to show that microbial composition at age one year is associated with cognitive outcomes at age two [187]. The results revealed that children with high levels of *Bacteroides* (which is typical of infantile microbiota) exhibited the highest level of performance on the Mullen Scales of Learning. Whereas, high levels of *Faecalibacterium* (typically associated with adult microbiota) were associated with lower performance levels. Unequivocally, maintaining a healthy gut during the first years of life is crucial for future cognitive development: Early prolonged states of enteric infections and early malnutrition have been linked to liver dysfunction and dysbiosis which synergistically exert systemic and homeostatic effects. Specifically, the subsequent microbial imbalances may cause leaky gut syndrome facilitating pathogenic bacterial translocation and, as such, promoting inflammatory responses and microglia activation, which in turn, can affect the infant’s BBB permeability and white matter integrity, and may ultimately lead to long-lasting cognitive and behavioral defects [188]. In addition to infections and malnutrition, several other factors have been shown to influence early microbiota development and diversity ranging from the delivery partum, infant feeding modes (e.g., infants fed with formula present with increased pathogenic populations of *Citrobacter*, *Enterobacter*, *Bilophla*, and *Ganulicatella*, while breast-fed infants have a higher prevalence of *Lactobacillus* and *Bifidobacteria* [189]), environmental exposures (including siblings and pets) and antibiotics consumption to host genetics [190]. GMC also varies as a function of regional diet, with significant consequences for healthy development. For example, when comparing African and Western diets, African and Finnish infants exhibited distinct microbiota being dominated by *Bacteroides* and *Prevotella* versus *Clostridium* and *Staphylococcus aureus*, respectively [191]. 

### 3.3. The Impact of Pro- and Prebiotics through the GBA

One way of influencing microbial diversity and reversing dysbiosis is via nutrition and drastic changes in diet have been found to alter microbial diversity in mere days [192]. Recent research suggests that microbial ecology can be therapeutically modified via the intake of so-called ‘psychobiotics’ [193], referring to active compounds that are capable of acting on the nervous system, consequentially shaping psychological processes and behavior, ultimately exerting health benefits in persons with psychiatric conditions. Indeed, psychobiotics have anxiolytic and antidepressant effects marked by changes in emotional, cognitive, systemic and neural indices [193]. Initially, only probiotics were considered as psychobiotics since they have the ability to release neuroactive substances (depending on the strain of belonging) [194]; this includes, for example, the production of dopamine and noradrenaline by members of the *Bacillus* genus, GABA by the *Bifidobacteria* genus, serotonin by the *Enterococcus* and *Streptococcus* genera, noradrenaline and serotonin by the *Escherichia* genus, and GABA and acetylcholine by the *Lactobacilli* genus [193]. Researchers have consistently outlined the psychotropic effects of probiotics: in an animal study, Barrett and colleagues demonstrated that *Lactobacillus (L.) brevis* and *Bifidobacterium (B.) dentium* increased GABA concentrations in vitro [195]. These findings were supported by a study using an in vivo mouse model showing that ingestion of the *L. rhamnosus* strain regulated emotional behavior and central GABA receptor expression [196]. Mice fed with *L. rhamnosus* exhibited decreased GABA mRNA expression in the amygdala, along with lower levels of stress-induced corticosterone and reduced anxiety- and depression-related behavior [196]. Another study replicated these results [197], and showed that the increase in GABA metabolites in the brain was evident after about four weeks, a lag that is comparable with the onset of other pharmaceutical interventions, such as serotonin-reuptake inhibitors [198]. Importantly, it was pointed out that re-colonization of gut bacteria in adolescent GF mice was not sufficient to reverse anxiety-like behavior, further supporting the idea that early deficits in gut microbiota may not be reversible.

With regards to human studies, Pärtty and colleagues administered *L. rhamnosus* to seventy-five infants aged six months and followed up with them for 13 years. The results suggested that supplementation with this probiotic strain may reduce the incidence of ADHD and Autism Spectrum disorders. However, the underlying mechanisms of action need further clarification, as no consistent microbial patterns could be identified [199]. In a randomized double-blind trial including 55 adult participants, it was found that the consumption of probiotics led to reduced measures of mood and distress, as well as decreased levels of urinary free cortisol, reflecting a decreased stress response [200]. Similarly, the results of a placebo-controlled, four-week probiotic food-supplement intervention study with multispecies probiotics in 20 healthy participants without mood disorders revealed a significantly reduced overall cognitive reactivity to sad mood (assessed by the revised Leiden index of depression sensitivity scale) as compared to the placebo group [201]. Lastly, consumption of a fermented milk product with probiotic for a period of four weeks by healthy women resulted in altered activity of brain regions involved in the control of emotion processing and regulation [202]. 

In addition to probiotics, prebiotics have more recently been classified as psychobiotics. Prebiotics are specific non-digestible food components, which selectively feed beneficial gut bacteria, consequentially stimulating their growth and activity with remarkable effects on brain development and function [193]. Prebiotics include oligosaccharides, fructans, unsaturated fatty acids, polyphenols and dietary fibers. To date, fructooligosaccharides (FOS) and galactooligosaccharides (GOS) have been studied the most, showing promising effects in animal models [203,204]. For example, milk oligosaccharides administration has been shown to prevent stress-induced dysbiosis and anxiety-like behavior in mice. Similarly, chronic combined FOS and GOS supplementation exhibited anxiolytic and antidepressant effects, as well as a reduction in the corticosterone stress response in mice [204]. In addition, prebiotics have been shown to modulate hippocampal and hypothalamus gene expression, and induce changes in SCFA concentration, which positively correlate with the behavioral effects. Further supporting the beneficial impact of prebiotics, a recent study in mice demonstrated that combined FOS-GOS supplementation from birth was associated with reduced anxiety-like and improved social behavior. Importantly, supplementation of short-chain GOS and long-chain FOS also affected serotonergic brain network regions comprising the prefrontal cortex (PFC) and the somatosensory cortex, and increased BDNF mRNA expression in the PFC [205]. In infants 12 months of age, administration of a combination of *B. longum* (BL999), *L. rhamnosus*, inulin, fructo-oligosaccharides (FOS), and LCPUFAs for one year resulted in higher, albeit not significantly different, scores in cognition and adaptive behavior [206]. More in-depth research on the potential beneficial effects of psychobiotics within such a critical developmental time window is needed, especially in light of the promising findings in older subjects. Therefore, an increase in processing of positive versus negative attentional vigilance and a significantly lower cortisol awakening response were observed in healthy adults after 3 weeks daily B-GOS intake, compared to the placebo group [207]. In a very recent study, the effects of prebiotic intake for four weeks on psychological and behavioral emotion regulation trait indices and the underlying brain networks involved were investigated in 60 girls in late adolescence [208]. The results showed a significant decrease in self-reported anxiety levels in the prebiotic group, along with a change in overt emotional processing in the dot-probe task. Moreover, the analyses of the pre- and postintervention stool samples showed a significant increase in beneficial *Bifidobacteria* in the gut microbiome. Together, these results suggest that four weeks of prebiotic intake is sufficient to induce changes in the microbial composition that lead to reduced anxiety levels in late adolescence. 

Psychobiotics can also act on the brain via modification of metabolic dynamics. For example, they can modulate tryptophan availability [209,210], which might have an impact on the kynurenine pathway that is responsible for 90% of tryptophan metabolization [113]. The downstream metabolites -kynurenic acid and quinolinic acid- of this pathway have recently been identified as relevant for the nervous system, as they exert neuroprotective, and excitotoxic effects, respectively, through their interaction with N-methyl-D-aspartate (NMDA) receptors [211,212]. Notably, research has suggested that dysfunctions of NMDA receptors during early development might cause CNS disorders, such as autism spectrum disorder (ASD) and attention deficit disorder (ADD), later in life [213,214]. This further emphasizes the importance of proper nutrition early in life, and indicates that psychobiotic supplementation might be an effective approach to influence tryptophan-kynurenine metabolism and prevent atypical developmental trajectories. In conclusion, we propose that in addition to the more traditionally recognized nutrients discussed in this review, emerging nutrients such as polar lipids, high quality protein/specific amino acids, and psychobiotics, are of critical importance for normal neurodevelopment in young children (see Figure 2, adapted from [215]).

## 4. Conclusions and Future Perspectives

Providing adequate amounts of all macro- and micronutrients in early life is essential for normal brain and neurocognitive development, yet based on the review, presented here, it is highly unlikely that any one nutrient alone represents a magic bullet in supporting and positively influencing neurodevelopment. This is simply not biologically plausible given the plethora of bioactive factors in human nutrition that may affect neurodevelopment. The human body is a host to other variables that may account for differences in developmental outcomes between formula and breastfed infants. For example, maternal IQ, household socio-economic status, and home nurturing are well-known confounders. Therefore, a number of questions remain to be answered, including the optimal nutritional composition for maximum development support, the optimal timing for nutritional interventions and whether boys and girls require different nutritional support. Polar lipids and essential amino acids such as tyrosine and tryptophan (high quality—available-protein), are emerging as promising emerging nutritional vectors for support of normal neurodevelopment. The latter can be especially important under conditions of protein/energy malnutrition. Moreover, the GBA represents a novel target for influencing neurodevelopment in young children and supplementation with psychobiotics (e.g., pre- and probiotics), in order to affect GBA dynamics is a promising emerging approach to promote cognitive, emotional, and neural development. More nutritional trials (observational and interventional) are now required to validate whether the nutrients and mechanisms described in this manuscript have practical significance and are able to induce clinically relevant immediate or long-term effects on health and neurocognitive developmental outcomes in healthy young children. 

## Figures and Tables

**Figure 1 nutrients-13-00199-f001:**
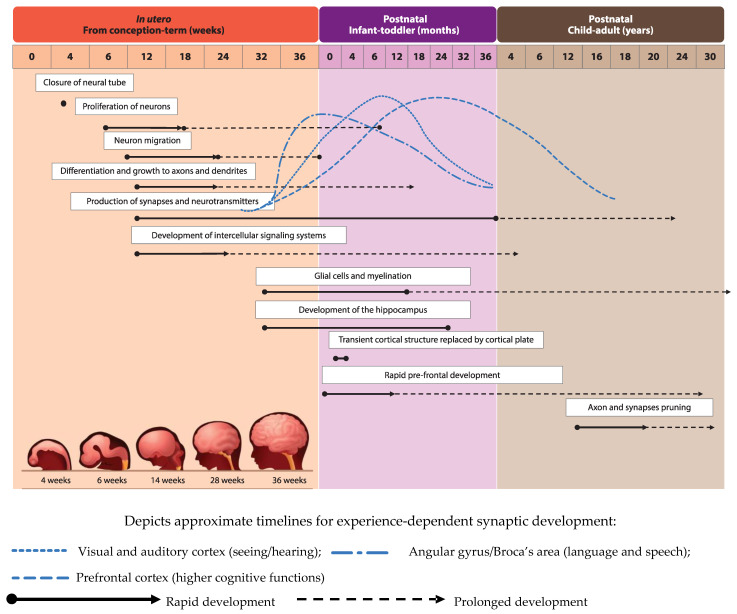
Visual representation of brain development timeline in humans from in utero up to adulthood.

**Figure 2 nutrients-13-00199-f002:**
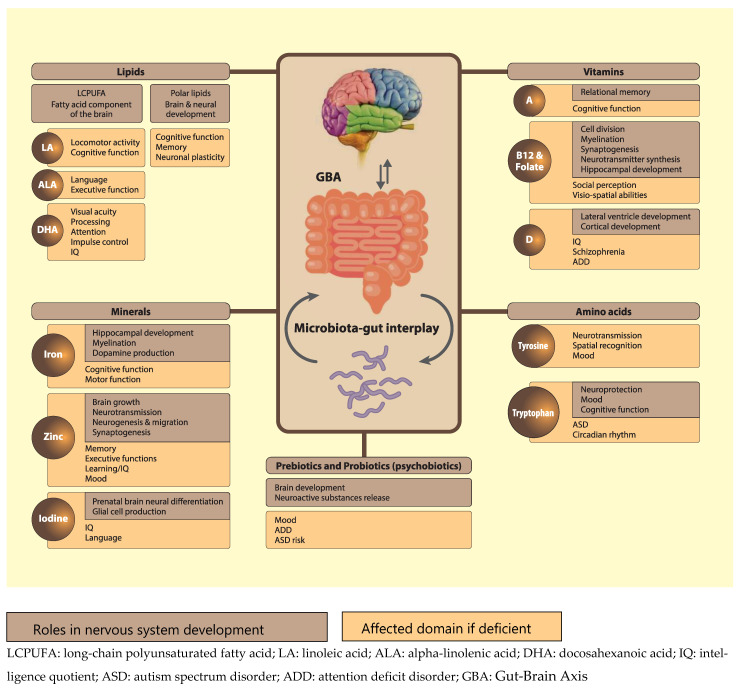
Functions and effect of some nutrients on brain and neuronal development. It also includes pre-and probiotics and tryptophan-based interactions through the gut brain axis.

## Data Availability

Detailed information is available upon request.

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
