# Peer review of "Nutritional Support of Neurodevelopment and Cognitive Function in Infants and Young Children—An Update and Novel Insights"

_nutrients, 2021, doi:10.3390/nu13010199_

Round 1

Reviewer 1 Report

This manuscript provides a systemic review for the function of several nutrients in brain development. Suggestions to improve the review include:

  1. Combine the “polar lipids” with the “long chain PUFA” sections together as one lipid section.
  2. The discussion regarding the effect of iron needs to be extended.
  3. The “Vitamins” section appears to be less organized. Different vitamins could be listed as sub-titles and discussed separately.
  4. The “protein and amino acids” part could be improved by adding subtitles to organize the discussion.
  5. Cellular mechanisms underlying the gut-brain axis could be further expanded in the discussion.

Author Response

  1. Reviewer’s comment: Combine the “polar lipids” with the “long chain PUFA” sections together as one lipid section.

We agree with the reviewer and combined the “polar lipids” section with the other lipid sections. The numbering of the various sections was adapted accordingly.

  1. Reviewer’s comment: The discussion regarding the effect of iron needs to be extended.

The discussion on effect of iron has been extended by adding: “Positive impact of iron supplementation on cognitive function seems to be observed only in anemic primary school children (Larson et al., 2017; Low et al., 2016). Based on the available evidence, adequate dietary iron intake should be encouraged during pregnancy and post-natal life up to adulthood. With regard to iron supplementation, a different picture emerges. Given the uncertainty of the efficacy of iron supplementation due to significant supplementation heterogeneity across the various studies (i.e. type and format of iron supplementation, dosage, length of supplementation, presence of other nutrients) (Larson et al., 2017; Szajewska et al., 2010) combined with the potential negative impact of providing high iron dosages to iron-replete infants (Lozoff et al., 2012), it seems advisable to restrict the provision of iron supplements, in the right form and dosage, to anemic individuals.”

One additional citation was included (Low et al 2016).

  1. Reviewer’s comment: The “Vitamins” section appears to be less organized. Different vitamins could be listed as sub-titles and discussed separately.

Thank you for this suggestion. Subsections 2.3.1-2.3.3 have been added to further organize this section.

  1. Reviewer’s comment: The “protein and amino acids” part could be improved by adding subtitles to organize the discussion.

The following subtitles have been introduced to improve readability of this section:

2.4.1. The importance of protein quality

2.4.2. mTORC1: linking protein (in)adequacy to brain development

2.4.3. Tryptophan

2.4.4. Tyrosine and Phenylalanine

2.4.5. Branched chain amino acids

  1. Reviewer’s comment: Cellular mechanisms underlying the gut-brain axis could be further expanded in the discussion.

Thank you for this suggestion, we have now expanded the relevant sections to provide more information. Specifically, the following changes were made:

The original text in section 3.1. has been changed to “Further, PGN can cross the blood-brain barrier (BBB) and interact with PGN-sensing molecules and transporters. It has been suggested that the influence of gut microbiota might even start prenatally, by contributing to BBB integrity regulation (Braniste et al., 2014).”

“Interestingly, these age-dependent increases in PGN concentrations were found across different brain regions, such as the prefrontal cortex, the striatum and the cerebellum, which suggests that these mechanisms are domain-general and affect the entire developing brain.” has been added to section 3.1.

The first sentence of section 3.2. has been changed to “Evidence from animal models has shown that the gut microbiome not only influences initial brain development, including synaptogenesis and myelination of brain areas supporting motor functions and cognitive abilities (Diaz Heijtz, 2016), but that it also affects brain responsiveness and function across the entire life-span by regulating neurotransmitter, synaptic, and neurotrophic signaling systems and neurogenesis (Luczynski et al., 2016)”.

Reviewer 2 Report

Kathrin Coen Kadosh with colleagues wrote an excellent paper that represents an overview of the nutritional support of neurodevelopment and cognitive function in infants and children. The authors have analyzed a large amount of literature data which will make this paper novel and interesting for readers. In my opinion, all sections are designed and written very well.
Minor comment:
1) Please, provide better text resolution of figure legends (figure1, figure2).

Author Response

The resolution has been increased for both figures from 300 dpi to 600 dpi and font size of legends text is changed from 10 points to 12 points.